# Convergent epigenetic evolution drives relapse in acute myeloid leukemia

Kevin Nuno[1,2,3,4†], Armon Azizi[2,3,4,5†], Thomas Koehnke[2,3,4], Caleb Lareau[6,7], Asiri Ediriwickrema[1,2,3,4], M Ryan Corces[1,2,3,4,8,9,10], Ansuman T Satpathy[6,7,11,12], Ravindra Majeti[2,3,4*]

[1]Cancer Biology Graduate Program, Stanford University School of Medicine, Stanford, United States; [2]Institute for Stem Cell Biology and Regenerative Medicine, Stanford University School of Medicine, Stanford, United States; [3]Cancer Institute, Stanford University School of Medicine, Stanford, United States; [4]Department of Medicine, Division of Hematology, Stanford University School of Medicine, Stanford, United States; [5]University of California Irvine School of Medicine, Irvine, United States; [6]Department of Pathology, Stanford University, Stanford, United States; [7]Program in Immunology, Stanford University, Stanford, United States; [8]Gladstone Institute of Neurological Disease, San Francisco, United States; [9]Gladstone Institute of Data Science and Biotechnology, San Francisco, United States; [10]Department of Neurology, University of California, San Francisco, San Francisco, United States; [11]Parker Institute for Cancer Immunotherapy, Stanford University, Stanford, United States; [12]Gladstone-UCSF Institute of Genomic Immunology, San Francisco, United States

*For correspondence:
rmajeti@stanford.edu

†These authors contributed equally to this work

**Abstract** Relapse of acute myeloid leukemia (AML) is highly aggressive and often treatment refractory. We analyzed previously published AML relapse cohorts and found that 40% of relapses occur without changes in driver mutations, suggesting that non-genetic mechanisms drive relapse in a large proportion of cases. We therefore characterized epigenetic patterns of AML relapse using 26 matched diagnosis-relapse samples with ATAC-seq. This analysis identified a relapse-specific chromatin accessibility signature for mutationally stable AML, suggesting that AML undergoes epigenetic evolution at relapse independent of mutational changes. Analysis of leukemia stem cell (LSC) chromatin changes at relapse indicated that this leukemic compartment underwent significantly less epigenetic evolution than non-LSCs, while epigenetic changes in non-LSCs reflected overall evolution of the bulk leukemia. Finally, we used single-cell ATAC-seq paired with mitochondrial sequencing (mtscATAC) to map clones from diagnosis into relapse along with their epigenetic features. We found that distinct mitochondrially-defined clones exhibit more similar chromatin accessibility at relapse relative to diagnosis, demonstrating convergent epigenetic evolution in relapsed AML. These results demonstrate that epigenetic evolution is a feature of relapsed AML and that convergent epigenetic evolution can occur following treatment with induction chemotherapy.

## Editor's evaluation

The authors show convincingly that most relapses in AML occur without changes in driver mutations. By using ATAC-seq in matched diagnosis and relapsed samples, they show that epigeneticc mechanisms drive relapse in a large proportion of cases. These findings are of translational importance and are based on rigorous analysis of large set of primary samples.

**eLife digest** Acute myeloid leukemia (or AML for short) is a type of blood cancer characterized by abnormally high production of immature white blood cells. Despite advances in AML treatment, many patients relapse after an initially successful first round of treatment. As a result, understanding the factors contributing to relapse is essential for developing effective treatments for the disease.

Like most cancers, AML can evolve because of changes to the DNA sequence in cells that cause them to grow uncontrollably or resist treatment. Alongside these genetic mutations, AML cells also undergo 'epigenetic' changes, where regions of the DNA are modified and genes can be switched on or off without altering the DNA sequence. Previous research has demonstrated that epigenetic changes contribute to the development of AML, however, it was not clear if these changes could also make cells resistant to treatment without acquiring new DNA mutations.

Nuno, Azizi et al. addressed this question by analyzing the epigenetic states of AML cells from 26 patients at the time of their diagnosis and after treatment when the disease had relapsed. Analysis revealed that almost half of the patients with AML experienced a relapse without acquiring new DNA mutations. Instead, these AML cells developed specific epigenetic changes that helped them to resist cancer treatment. Moreover, studying individual AML cells from different patients showed that the cells became more epigenetically similar at relapse, suggesting that they converge towards a more treatment-resistant disease.

Future experiments will determine exactly how these epigenetic changes lead to treatment resistance. Currently, most of the drugs used to treat AML are either chemotherapies or ones that target specific DNA mutations. The findings of Nuno, Azizi et al. suggest that drugs targeting specific epigenetic changes may be more effective for some patients. Further studies will be needed to determine which patients may benefit and which epigenetic drugs could be useful.

## Introduction

Acute myeloid leukemia (AML) is a blood cancer characterized by the accumulation of dysfunctional myeloid progenitors, resulting in severe cytopenias and an overall poor prognosis, especially in the elderly (*Döhner et al., 2015*). Despite recent advances in AML treatment, most patients relapse with aggressive disease that is highly resistant to further treatment. Relapse therefore presents a significant challenge for AML clinicians due to the lack of efficacious salvage options making it critical to study AML relapse to guide and improve outcomes.

Cancer progression is generally understood as an evolutionary process with certain cellular subpopulations containing genetic and other features that allow for improved fitness and growth. There is often substantial subclonal genetic heterogeneity in cancer, with a dominant clone comprising the bulk of the tumor with accompanying smaller subclones (*Akbani et al., 2015*; *Koboldt, 2012*; *The Cancer Genome Atlas Network, 2012*; *Ding et al., 2012*). Leukemia can persist after treatment during clinical remission as measurable residual disease (MRD) which is an independent risk factor for relapse (*Ding et al., 2012*). Indeed, whole-genome resequencing efforts have indicated that in many cases, clonal evolution at relapse results from the dynamic outgrowth of these resistant clones (*Hassan et al., 2017*). These data underscore the importance of characterizing the clonal dynamics of AML evolution to better understand how these cells evolve over time and contribute to relapse and chemoresistance.

According to the cancer stem cell model, AML is thought to be organized in a hierarchical manner, with transformed leukemia stem cells (LSCs) possessing long-term self-renewal capability giving rise to partially differentiated leukemia cells (non-LSCs; *Dick, 2005*; *Thomas and Majeti, 2017*). The role of LSCs in relapse is poorly understood, and future studies seeking to understand relapsed AML will require investigating how these cells contribute to disease recurrence. These studies are especially important given the lack of therapies that specifically target LSCs.

Recent genomic sequencing efforts have identified various classes of recurrent mutations in AML (*The Cancer Genome Atlas Research Network, 2013*; *Tyner et al., 2018*). In particular, these studies implicate epigenetic regulators as key factors in AML pathogenesis, including DNMT3A, TET2, IDH1/2, ASXL1, and the cohesin complex (*Chan and Majeti, 2013*; *Ley et al., 2010*; *Delhommeau et al., 2009*; *Gross et al., 2010*; *Abdel-Wahab et al., 2012*; *Kon et al., 2013*). Indeed, epigenetic

dysregulation is now known to be a distinctive hallmark of AML (*Hu and Shilatifard, 2016*). Studies analyzing the transformation of normal hematopoietic stem cells (HSCs) into LSCs suggest epigenetic dysregulation is an initiating event in leukemic clonal evolution, and many mutations in epigenetic regulators persist between diagnosis and relapse (*Jan et al., 2012*; *Corces-Zimmerman et al., 2014*; *Shlush et al., 2014*).

It is increasingly apparent that chromatin accessibility and its three-dimensional landscape are important factors in cancer pathogenesis and progression (*Corces and Corces, 2016*). Recent studies profiling the chromatin landscape of certain cancers indicate that the fine regulation of chromatin activity is critical for tumor suppression (*Hnisz et al., 2016*; *Flavahan et al., 2016*). Analyses of active epigenetic programs in healthy hematopoietic and leukemic cells implicate epigenetic factors, including distal *cis*-regulatory elements and transcription factors, as critical features driving AML, with differential activation of regulatory chromatin programs defining disease subtypes (*Corces et al., 2016*; *McKeown et al., 2017*). Moreover, common AML mutations in transcription factors and RAS/RTK signaling have downstream activity on trans regulatory factors like NFkB, highlighting the importance of epigenetic regulatory elements in maintaining the balance of transcriptional programs required for proper cellular identity and preventing leukemic transformation (*Corces et al., 2018*; *Assi et al., 2019*). Cytosine methylation states have also been shown to evolve independently of genetic mutations in relapsed AML, indicating that epigenetic changes can occur with leukemia progression (*Li et al., 2016*). We hypothesize that examining chromatin accessibility to infer cancer gene regulatory programs and cancer-specific properties is a useful tool for uncovering novel biology in AML progression and relapse (*Rosenbauer and Tenen, 2007*).

Here, we report a longitudinal study of paired diagnosis-relapse AML patient samples to characterize mechanisms of AML relapse. We first interrogated previously published genotyping studies of relapsed AML to understand clonal evolution of somatic driver mutations and found a large proportion of cases lack mutational changes at relapse, suggesting that epigenetic evolution could be a driving factor in relapse. We then performed chromatin accessibility analysis to identify regulatory mechanisms associated with AML relapse in mutationally stable cases and further performed single-cell chromatin analysis to investigate epigenetic subclones and evolution in primary patient samples. These studies reveal that individual diagnosis clones persist and acquire similar epigenetic features and states in relapse, demonstrating convergent epigenetic evolution.

## Results
### A large proportion of relapsed AMLs lack mutational evolution

To define the genetic landscape underlying AML relapse, we analyzed published genotyping studies of paired diagnosis and relapse samples. We collected data from seven different cohorts with genome resequencing of AML-related mutations (*Supplementary file 1a*; *Tyner et al., 2018*; *Li et al., 2016*; *Greif et al., 2018*; *Shlush et al., 2017*; *Parkin et al., 2013*; *Rothenberg-Thurley et al., 2018*; *Stratmann et al., 2021*). For each patient, we collected mutation variant allele frequency (VAF), karyotype, time to relapse, and overall survival when available. This resulted in a dataset containing 2111 mutations across 216 patients at paired diagnosis and relapse timepoints.

Recurrently mutated genes within this cohort were consistent with previous studies with NPM1, DNMT3A, FLT3, RUNX1, and TET2 being among the most frequently mutated genes (*Figure 1a*). We previously showed that genes associated with epigenetic regulation such as DNMT3A, TET2, IDH1/2, and ASLX1 are pre-leukemic mutations that accumulate in HSCs and initiate leukemogenesis (*Jan et al., 2012*; *Corces-Zimmerman et al., 2014*). The VAF of mutations in epigenetic regulators was overall stable at relapse, consistent with the idea that these mutations are founder events occurring early during AML evolution and suggesting that the persistence of these mutations may play a role in AML resistance and progression (*Figure 1b*, *Figure 1—figure supplement 1a*). FLT3 and other signaling molecules such as those in the Ras family, including KRAS, NRAS, and PTPN11, were frequently lost at relapse (*Figure 1b*, *Figure 1—figure supplement 1b*). Mutations in WT1 were gained at relapse in 10.8% of cases in this study, a frequency greater than any other gene evaluated. Patients with WT1 gain took significantly longer to relapse relative to patients that had WT1 mutations already present at diagnosis (*Figure 1—figure supplements 1c and 2a*).

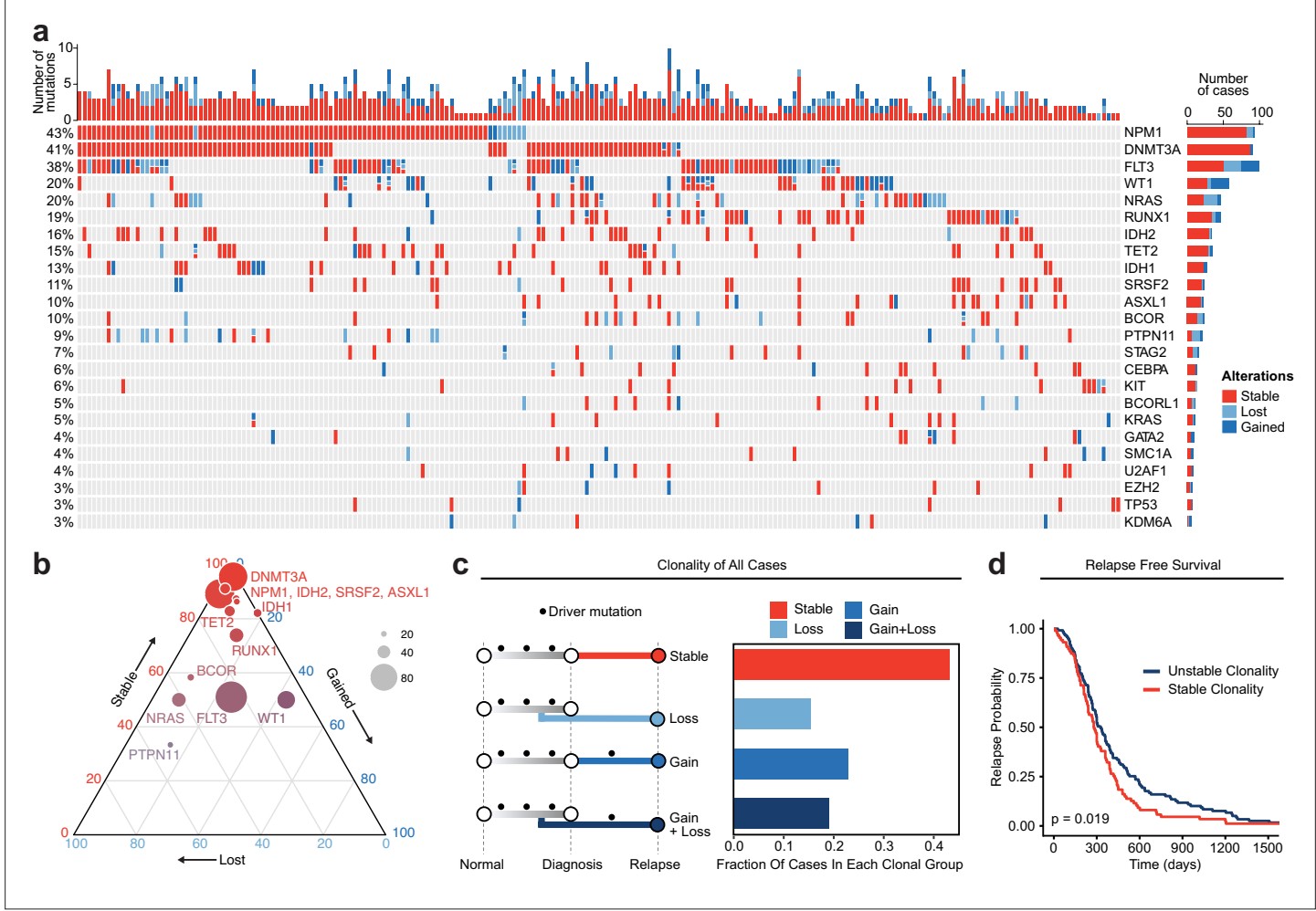

**Figure 1.** Meta-analysis of genomic change in relapsed AML. (**a**) Oncoprint of recurrently mutated genes from published relapsed AML dataset. Genetic alterations are color coded according to dynamics between disease state: stable at relapse, gained at relapse, or lost at relapse. The number of gained, lost, and stable mutations as well as percentage of patients with mutations in each gene are displayed on the right and left sides of the plot respectively. (**b**) Ternary plot of recurrently mutated genes from dataset analysis depicting mutational dynamics between diagnosis and relapse (restricted to genes with at least 20 events in our cohort). The sides of the plot correspond to the percentage of mutations displaying the indicated dynamic (gained, lost, or stable at relapse), the size of each dot indicates the number of mutations detected for that gene. (**c**) Schematic of clonal 'bins' each patient was categorized into based on mutational data. Bar chart on the right depicts the proportion of patients assigned to each clonal bin. (**d**) Survival analysis of patients based on sample clonality data gathered for each patient group. 'Stable' clonality refers to patients with equivalent mutations detected at diagnosis and relapse, 'Unstable' refers to patients that had any change in mutations at relapse (log-rank test p<0.05).

The online version of this article includes the following figure supplement(s) for figure 1:

**Figure supplement 1.** Plots depicting change in variant allele frequency for detected 'pre-leukemic' epigenetic modifier mutations (**a**), RAS pathway mutations (**b**), or WT1 mutations (**c**) in meta-analysis relapsed AML cohort.

**Figure supplement 2.** Gene-specific clonal analysis and multivariate analysis of relapse-free-survival based on clonality.

To evaluate genetic clonal patterns at relapse, we utilized the available genomic VAF data with standardized cutoffs for detection, acquisition, and elimination of mutations to infer the acquisition and elimination of clones at relapse. Each mutation was categorized as (1) gained (VAF from <0.05 at diagnosis to >0.1 at relapse), (2) lost (VAF from >0.1 at diagnosis to <0.05 at relapse), or (3) stable (not meeting criteria for gained or lost) at relapse. Each patient was categorized into one of four different bins based on the AML driver mutations detected at diagnosis and relapse: stable (no change in mutations between diagnosis and relapse), gain (acquisition of mutations at relapse); loss (loss of mutations at relapse); and gain and loss (both gain and loss of mutations at relapse) (*Figure 1c*). Notably, 93 cases (43%) exhibited stable mutation profiles at relapse. There were no significant differences in

these clonal group frequencies across individual genes, and there was no clear enrichment of specific epigenetic mutations such as DNMT3A, TET2, or IDH1/2 in our stable samples, indicating that clonal evolution categories are unlikely to be driven by or associated with specific mutations (*Figure 1—figure supplement 2b*). Patients with stable clonality had a significantly shorter time to relapse relative to patients who gained and/or lost mutations at relapse when corrected for differences in age and sex (*Figure 1d*, *Figure 1—figure supplement 2c*; HR = 1.56, p=0.007). These data indicate that little to no somatic genetic evolution occurs in driver mutations in a substantial portion of relapsed AML cases and that these cases are associated with a shorter time to relapse.

## Epigenetic evolution is an important non-genetic factor in AML relapse

The identification of many clonally stable relapsed patients led us to hypothesize that non-genetic or epigenetic factors may contribute to relapse in these cases. To evaluate this hypothesis, we identified 26 AML patients treated at Stanford with available banked diagnosis and relapse samples. The majority of these patients were treated with induction chemotherapy consisting of cytarabine with anthracycline and all patients entered clinical remission prior to eventually relapsing (*Supplementary file 1b*). We FACS-purified bulk AML blasts from each of these patients to exclude residual lymphocytes and subjected these cells to ATAC-seq and genotyping with a targeted panel of AML-associated genes (Blast immunophenotype: CD45-mid SSC-hi non-CD34 +CD38-, *Figure 2a*). We used a custom genotyping pipeline to identify pathogenic mutations occurring at VAF >5% (see Methods) and incorporated available cytogenetic data to assign each patient to a clonal group (*Supplementary file 1c and d*). We observed a similar pattern of frequently mutated genes in this patient cohort, with NPM1, FLT3, DNMT3A, and TET2 occurring most frequently (*Figure 2b*). We classified each sample into clonal categories as described above and strikingly, we also observed a clonal pattern with a similar proportion of samples in each of the clonal bins defined above, with roughly 40% of our cohort exhibiting no mutation changes at relapse indicating that they were clonally stable (*Figure 2c*). Of note, our VAF cutoff and classification accurately classified all mutations as stable, gained, or lost; all stable mutations had an absolute VAF change of <0.1 with the exception of one mutation which had a VAF change of 0.35. We observed that time-to-relapse in patients with stable clonal evolution was not significantly different compared to non-stable cases, although this may be limited by the small sample size (p=0.2; *Figure 2—figure supplement 1a*).

Chromatin accessibility has been extensively used to profile epigenetic gene regulatory landscapes and programs (*Corces et al., 2016*; *Granja et al., 2019*). We sought to identify such non-genetic factors playing a role in the evolution of the different genomic clonal evolution groups. As a readout for epigenetic similarity, we calculated the Pearson product moment correlation of ATAC-seq signal normalized for reads in peaks across all peaks between diagnosis and relapse timepoints for each patient and compared these values between all clonal bins (*Figure 2d*). The different clonal groups showed varying levels of epigenetic change at relapse. Notably, relapsed samples with stable clonal evolution were epigenetically dissimilar to their diagnosis counterparts suggesting that epigenetic evolution may drive relapse in these cases. In addition, samples with both gain and loss of mutations showed substantial epigenetic changes at relapse which may be attributable to mutational changes occurring in these samples and/or to epigenetic factors (Mean DX-REL correlation: Stable-0.914, Gain-0.929, Loss-0.950, Gain +Loss−0.802). Collectively, these data suggest that epigenetic evolution may drive relapse in AML, particularly in cases that do not acquire or lose mutations.

## Cell cycle and metabolic pathways of epigenetic evolution underlie relapse in mutationally stable AML

The clonally stable group exhibited substantial chromatin accessibility changes at relapse suggesting that epigenetic evolution may contribute to resistance in these cases. To identify the specific regulatory elements and pathways involved, we identified regions of chromatin accessibility that were enriched or depleted at relapse in mutationally stable AMLs. Of 115,551 features in the global peak set, 2373 peaks were enriched and 6145 peaks were depleted across all stable samples (*Figure 2—figure supplement 1b*). To identify chromatin changes localized to specific gene loci, we generated chromatin accessibility 'gene accessibility scores' across all samples. This approach has been shown to predict the relative RNA expression of individual genes (*Granja et al., 2019*; *Granja et al., 2021*). We therefore used these scores to identify differentially accessible genes across stable samples with

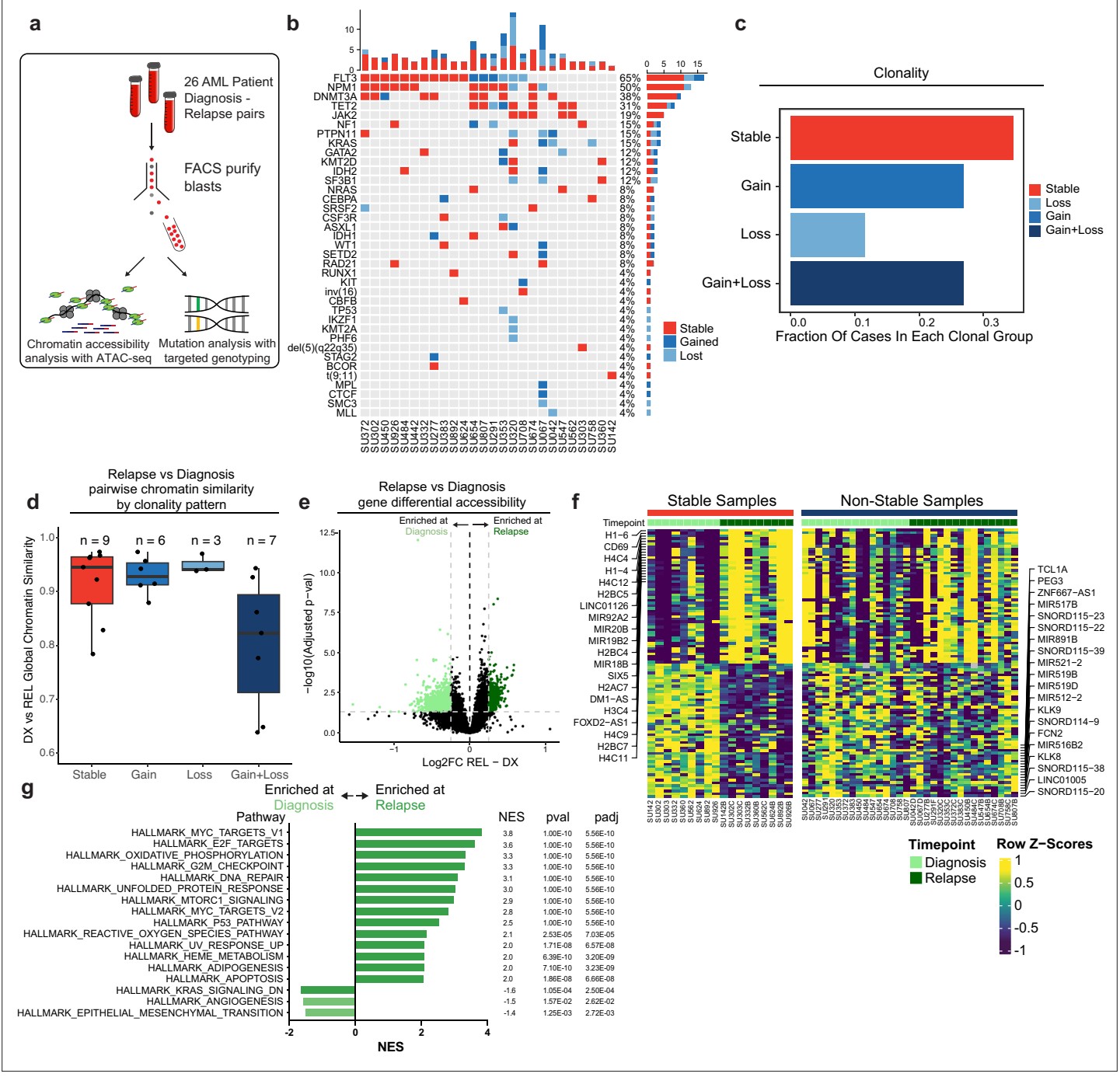

**Figure 2.** Chromatin accessibility change in stable and non-stable relapsed AML. (**a**) Schematic of relapse sample acquisition and preparation. (**b**) Oncoprint of recurrently mutated genes in samples analyzed from Stanford University patient cohort (n=26). (**c**) Bar chart depicting fractions of patients from Stanford patient cohort in each clonal bin. (**d**). Violin plot depicting global chromatin accessibility similarity of diagnosis/relapse pairs based on ATAC-seq data. Each dot represents a value calculated for each patient based on overall chromatin accessibility similarity between diagnosis and relapse samples. (**e**) Volcano plot depicting differentially accessible genes at relapse in patients within the Stable clonal bin. (**f**) Heatmap depicting the accessibility of the top differential genes at diagnosis and relapse in mutationally stable samples Left: Accessibility displayed for stable samples. Right: accessibility displayed for non-stable samples. (**g**). Gene set enrichment analysis (GSEA) of differentially open or closed accessible genes calculated with ATAC-seq gene scores in mutationally stable samples. Top enriched or de-enriched pathways crossing significance threshold of padj <0.05 are shown.

The online version of this article includes the following figure supplement(s) for figure 2:

**Figure supplement 1.** Relapse-free-survival and differential chromatin analysis of the Stanford AML cohort.

*Figure 2 continued on next page*

*Figure 2 continued*

**Figure supplement 2.** Projection of bulk AML ATAC samples to healthy hematopoietic single-cell manifold demonstrates shifts in epigenetic differentiation states at relapse.

the underlying hypothesis that differential chromatin accessibility may act as a proxy for transcriptomic changes in these samples (*Figure 2e and f*; *Granja et al., 2019*; *Granja et al., 2021*). We found that histone genes, epigenetic regulators, and the surface marker CD69 displayed increased accessibility at relapse across samples (*Figure 2f*, *Supplementary file 1e*). Differentially accessible genes in stable samples were not significantly differentially accessible in non-stable samples (AMLs that gained and/or lost mutations at relapse), indicating that mutationally stable samples undergo specific epigenetic evolution distinct from other AMLs (*Figure 2f*). We additionally evaluated whether relapse accessibility changes were specific to certain mutational groups based on our genotyping data. For the genes FLT3, TET2, DNMT3A, and NPM1 (genes with more than five mutant cases), we did not find any chromatin loci with evolution at relapse that was significantly different between mutant and wildtype cases.

To analyze the association between chromosomal regions and genes that were found to be differentially accessible across stable samples, we searched for chromosomal loci with a higher proportion of either significantly enriched or depleted peaks relative to the global peak set with the hypothesis that large regions of chromosomal accessibility change may represent copy number alterations or changes in topologically associated domain activity associated with relapse. We found that certain genomic loci displayed a widespread increase in average accessibility at relapse paired with increased accessibility of specific genes in those regions. This was in contrast to the rest of the genome where average accessibility and gene accessibility change at relapse were not correlated. For example, chromosome 6 contained a region of increased relapse accessibility spanning approximately 100 kb which contained numerous histone and cell-cycle-associated genes (*Figure 2—figure supplement 1c*). These locations may represent regions of co-regulated chromatin which evolve to allow for resistance and relapse.

We performed gene set enrichment analysis of genes with increased accessibility at relapse in stable samples and found enrichment of pathways associated with increased cell cycling, metabolism, and cell stress responses. These pathways were consistent with increased aggressiveness of relapsed AML and a response to genotoxic chemotherapy, as well as chromatin regulators, histones, and targets of transcription factors such as MYC (*Figure 2g*). We additionally examined transcription factor motif enrichment within differentially accessible chromatin peaks to identify *trans* regulatory factors important in mutationally stable relapse samples (*Figure 2—figure supplement 1d*). We found depletion of AP1 family motifs, such as FOS and JUN, which have been demonstrated to be important in various genetically defined subtypes of AML, as well as CEBPA and SPI1, both critical regulators of myeloid cell differentiation (*Assi et al., 2019*). Conversely, we observed an enrichment of FOXO family factors in the stable relapse samples, as well as the GATA family (known regulators of early progenitors in hematopoietic development) and E2F (*Rosenbauer and Tenen, 2007*). These results suggest differential transcription factor activity and activation of relapse-specific biological pathways in mutationally stable relapse cases.

We then utilized healthy hematopoietic cell type chromatin profiles to identify the closest normal cell type for AML samples. We previously used the CIBERSORT algorithm to quantify the cell type contribution to overall AML chromatin states (*Corces et al., 2016*). A follow-up study utilized latent semantic indexing projection of either single-cell ATAC or pseudo-single-cell ATAC (derived from bulk ATAC profiles) to healthy hematopoietic single-cell ATAC data to identify the closest normal cell type for single-cell ATAC profiles (*Granja et al., 2019*). We applied the latter, single-cell specific, methodology to our data in order to identify the closest normal cell types for diagnosis and relapse samples and by proxy, the differentiation status of these AMLs as determined by their chromatin state. Briefly, for each bulk AML sample, a 'pseudo-single cell' ATAC-seq dataset was generated as previously described (*Granja et al., 2019*). This pseudo-single cell data was then projected to a reference of healthy hematopoietic ATAC-seq data, and each pseudo-single cell was assigned the closest normal cell type using a k-nearest neighbors clustering approach *Figure 2—figure supplement 2a and b*. The closest normal cell types for diagnosis and relapse pairs were then compared across samples. This analysis revealed that at diagnosis, most AMLs mapped to

different regions within the myeloid/monocyte compartment and that at relapse, AMLs overall became less differentiated with chromatin reflecting a more CMP/GMP/progenitor-like cell state (*Figure 2—figure supplement 2b*). Evaluating individual cases showed that every sample which was myeloid/monocyte-like at diagnosis became less differentiated at relapse and that samples that were progenitor-like at diagnosis generally stayed within that compartment (*Figure 2—figure supplement 2c, d and e*). These analyses indicate that the overall chromatin state of relapsed AML becomes less differentiated, which may result in epigenetic fitness associated with drug resistance and increased proliferation.

## Epigenetic evolution at relapse is primarily driven by non-leukemic stem cell populations

Leukemia stem cells (LSCs) are thought to be of critical importance in AML relapse, serving as a chemoresistant self-renewing reservoir for disease propagation after initial therapy (*Thomas and Majeti, 2017*; *Jordan, 2007*). As LSCs increase in frequency at relapse, we sought to determine if the relapse-associated chromatin signature detected in the mutationally stable cases was reflective of an LSC epigenetic signature (*Ho et al., 2016*). We examined the chromatin accessibility profile and epigenetic evolution that occurs within populations enriched for LSCs, identified by expression of CD99 and TIM3 within the CD34 +CD38- population, markers that have been previously validated for distinguishing leukemic stem cells from non-leukemic residual HSCs (*Jan et al., 2011*). As CD34 expression can be variable in AML cases, we identified 10 patients within our cohort that contained a sortable CD34 +CD38 CD99 + TIM3 + population of cells in both diagnosis and relapse samples. We FACS-purified both this LSC-enriched population and the remaining non-LSC AML cells in these samples (designated 'mature non-LSCs') and subjected them to ATAC-seq (*Figure 3a*). The immuno-phenotype used to enrich LSCs is incompletely defined in the relapse setting, however, we found that that CD34 + CD38 CD99 + TIM3 + cells from relapse samples, mapped epigenetically to less-differentiated healthy cell states relative to non LSC-enriched samples, consistent with the hypothesis that the 'LSC' immunophenotype works effectively for enrichment in both diagnosis and relapse samples (*Figure 2—figure supplement 2b*).

Comparison of chromatin accessibility at diagnosis and relapse showed that LSCs underwent less global epigenetic change at relapse, indicating that most of the chromatin accessibility change at relapse occurs in the non-LSC compartment (*Figure 3b*). We next investigated how regions becoming differentially accessible in LSCs were related to chromatin changes in the non-LSC compartment and vice versa. This analysis revealed that almost half of the features with significant chromatin changes in LSCs were also differentially accessible in non-LSC cells (*Figure 3c and d*). However, fewer than 5% of features that were differentially accessible in non-LSCs were differentially accessible in the LSC compartment (*Figure 3c and d*). These results indicate that chromatin changes may be inherited in a directional manner within the AML hierarchy; chromatin changes in LSCs are passed on to their non-LSC progeny, but not vice versa. Overall, these results indicate that epigenetic changes detected in relapse samples are the result of chromatic accessibility alterations in both LSC and non-LSC compartments, rather than a change in the number or proportion of LSCs at relapse.

We aimed to determine whether epigenetic changes in the bulk AML at relapse (the bulk relapse signature identified in *Figure 2*) could be attributed to evolution of LSC-specific epigenetic features. We directly compared the gene accessibility signature identified in the stable relapse samples to an epigenetic signature derived by comparing LSCs to non-LSCs at diagnosis. The stable sample bulk AML relapse signature was strikingly similar to the LSC signature at diagnosis, indicating that a significant proportion of the epigenetic changes observed at relapse are due to changes in LSC frequency or evolution of non-LSC cells toward a more LSC-like state (*Figure 3—figure supplement 1A and B*).

To examine the LSC-specific pathways involving differentially accessible genes within these cells, we performed GSEA using our gene score data (*Figure 3e*). Enriched programs included many of those that were found in the bulk leukemia, including inflammatory pathways such as interferon-γ and IL-2/STAT5 signaling, as well as an upregulation of KRAS signaling pathway-related genes. Several pathways enriched in bulk leukemia at relapse, such as MYC target genes, were not enriched in LSCs at relapse, suggesting that LSCs undergo epigenetic alterations distinct from the bulk population (*Figure 3f*, leading edge genes depicted in *Figure 2f*). Overall, these results suggest that epigenetic evolution is more prominent in non-LSCs, that epigenetic change occurring in LSCs are reflected in

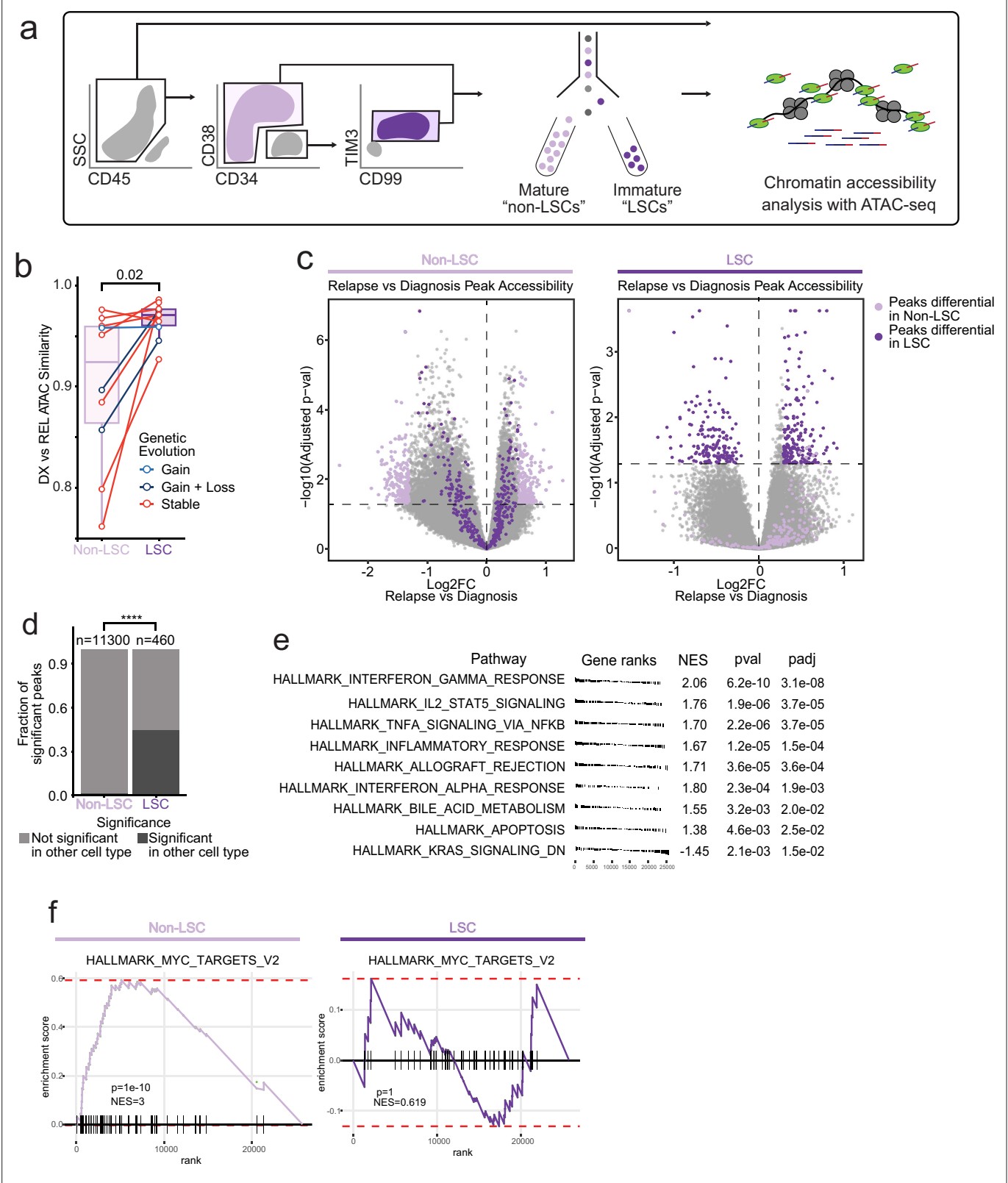

**Figure 3.** Chromatin change in relapsed AML LSCs. (**a**) Cell sorting and library preparation scheme for comparison of chromatin accessibility profiles of LSC vs. non-LSC enriched subpopulations in relapsed AML cohort. (**b**) Box and whisker plot depicting chromatin accessibility correlation coefficient for non-LSCs vs. LSCs. Lines connect LSC similarity and non-LSC similarity values for each patient. (**c**) Volcano plot of differentially accessible peaks between diagnosis and relapse in non-LSCs (left) or in LSCs (right). The top 200 differential peaks by log2FC in LSCs (dark purple) and non-LSCs (light

*Figure 3 continued on next page*

*Figure 3 continued*

purple) are highlighted in each plot. Cutoffs for adjusted p value of<0.05 are depicted as dashed lines in both plots. (**d**) Bar plot comparing fraction of significantly differentially accessible peaks in non-LSCs that were also differentially accessible in LSCs and vice versa. (**e**) Gene set enrichment analysis (GSEA) of differentially open or closed accessible genes calculated using gene scores from ATAC-seq data from LSC samples. A positive NES indicates that the pathway was upregulated at relapse relative to diagnosis based on the gene accessibility data. Top pathways crossing significance threshold of padj <0.05 are shown. (**f**) GSEA plot of exemplar gene pathway enriched in bulk Stable ATAC-seq samples (HALLMARK_MYC_TARGETS_V2) applied to either non-LSC gene score data set (left) or LSC gene score data set (right).

The online version of this article includes the following figure supplement(s) for figure 3:

**Figure supplement 1.** Comparison between AML relapse signature and LSC signature.

non-LSCs but not vice-versa, and that LSC epigenetic changes at relapse partially, but incompletely, reflect changes in the bulk leukemia cells.

## Single-cell ATAC-seq identifies epigenetically defined subpopulations at diagnosis and relapse

Several studies performed in AML, as well as glioblastoma and lung adenocarcinoma, have demonstrated intratumoral epigenetic heterogeneity that may contribute to disease progression and evolution (*Torres et al., 2016*; *Tavernari et al., 2021*). We sought to identify mechanisms of epigenetic evolution, particularly in cases lacking clonal somatic mutational changes at relapse, and hypothesized two different mechanisms: *intracellular* evolution, where the epigenome of individual cells changes to a relapse-specific state, or *intercellular* evolution, where evolutionary pressure selects for a subset of cells present at diagnosis that harbor a pre-existing relapse signature. To explore these possibilities, we performed 10 X single-cell ATAC-seq (scATAC-seq) on three diagnosis and relapse sample pairs, each of which contained distinct and varied AML-related genetic lesions (*Supplementary file 1b and c*), underwent varying degrees of epigenetic change at relapse according to our bulk analysis, and, importantly, exhibited no mutational clonal evolution. We additionally analyzed one sample pair which underwent epigenetic and genomic change at relapse. We employed previously validated computational tools to analyze epigenetic heterogeneity within and between samples (*Figure 4a*; *Corces et al., 2018*; *Granja et al., 2019*; *Granja et al., 2021*).

Dimensionality reduction analysis of selected single-cell ATAC samples revealed substantial inter- and intra-patient heterogeneity (*Figure 4b*). In patient SU142 (MLL-rearranged), diagnosis and relapse clusters were indistinguishable from each other, indicating that little epigenetic change occurred during relapse in this case (*Figure 4—figure supplement 1a*). In patients SU360 (SF3B1 and KMT2D mutant secondary AML), SU484 (IDH1/FLT3-ITD/NPM1 mutant), and SU892 (RUNX1-mutant/FLT3-TKD), there were substantial epigenetic differences between diagnosis and relapse, along with significant heterogeneity within each timepoint (*Figure 4c and d*, *Figure 4—figure supplement 1a*). Each sample and timepoint contained several epigenetically defined cell subpopulations, indicating that epigenetically defined clusters can exist independent of mutational differences (*Figure 4—figure supplement 1a*).

We hypothesized that in genetically stable AMLs, there may exist cells at diagnosis with epigenetics states more amenable to selection during therapy ultimately driving relapsed disease. To test this hypothesis, we calculated the epigenetic similarity between all diagnosis and relapse clusters by comparing global *chromatin* accessibility. Multiple samples contained discreet clusters at diagnosis which were epigenetically more similar to clusters at relapse relative to other cells at diagnosis (*Figure 4c and d*, *Figure 4—figure supplement 1b*). Specifically, in patient SU360, diagnosis clusters 7, 8, and 9, which represented 24% of the AML at diagnosis, grouped with several of the relapse clusters (*Figure 4c*). Similarly, for patient SU892, a sample that displayed more substantial global epigenetic change at relapse, we found that diagnosis clusters 1 and 2, representing 14% of the AML at diagnosis, were most similar to relapse cells (*Figure 4d*). These analyses indicate that there exists substantial epigenetic heterogeneity within AMLs at multiple therapeutic timepoints and that some diagnosis subpopulations contain epigenetic states similar to cells at relapse.

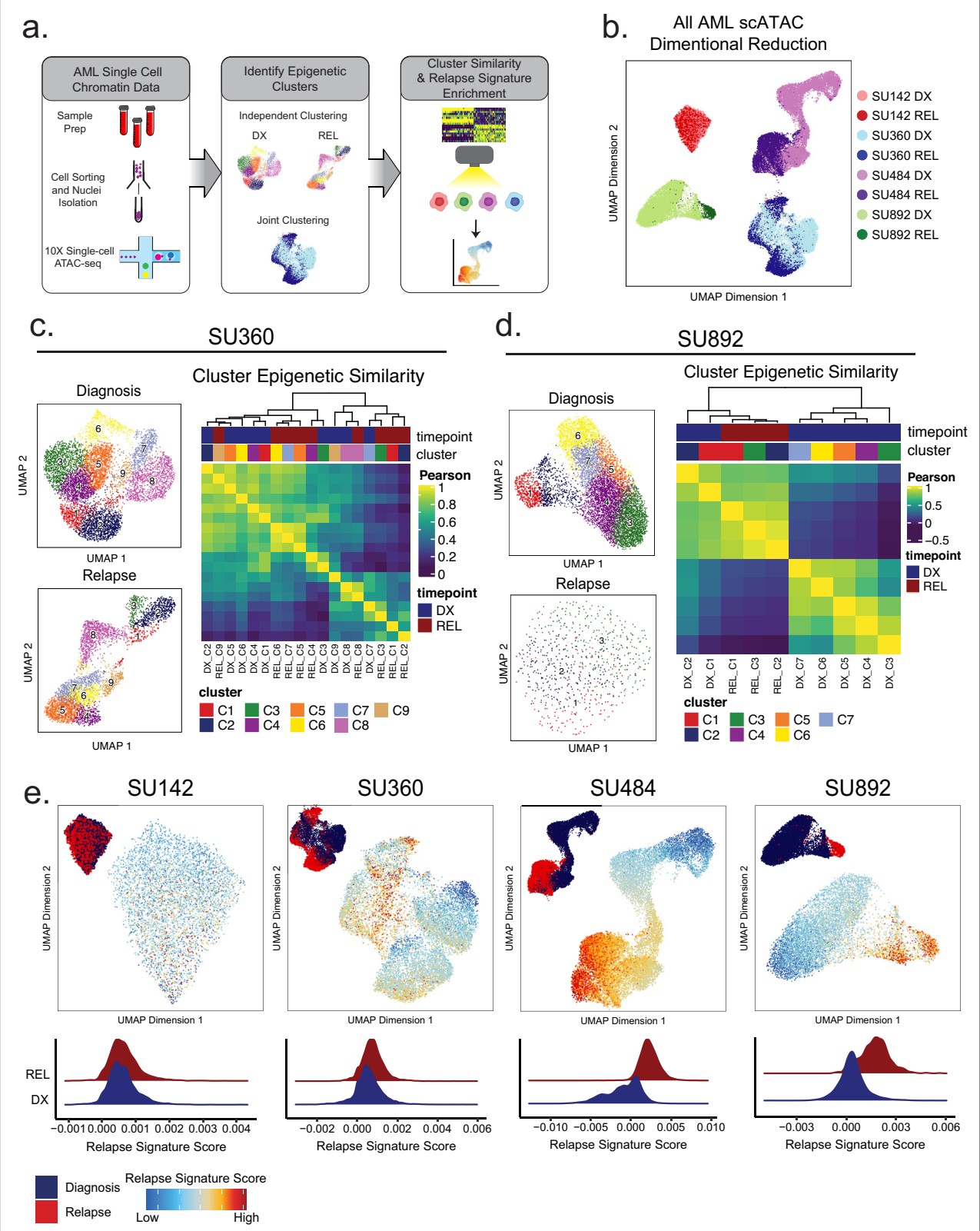

**Figure 4.** Single-cell ATACseq analysis of relapsed AML. (**a**) Scheme for single-cell ATAC-seq sample preparation and cell clustering analysis (**b**) UMAP projection of all single-cell ATAC-seq patient samples from Stanford diagnosis/relapse cohort (samples color-coded according to scheme at right). (**c–d**) Cell clustering analysis of Stanford patient samples SU360 (**c**) and SU892 (**d**). At left, UMAP plots of individual samples from each patient with color-coded cell clusters shown; at right, non-hierarchical clustering heatmap of determined cell clusters. (**e**) UMAP projections of single-cell ATAC-seq from

*Figure 4 continued on next page*

*Figure 4 continued*

patient samples showing heat signature for Stable sample relapse ATAC-seq signature (from *Figure 2*). Inset UMAP shows cells colored according to diagnosis or relapse samples. Overall relapse score for each sample is quantified in plots at the bottom.

The online version of this article includes the following figure supplement(s) for figure 4:

**Figure supplement 1.** Single-cell SNN clusters and between-cluster epigenetic similarities.

**Figure supplement 2.** Projection of AML scATAC samples to healthy hematopoietic single-cell manifold demonstrates shifts in epigenetic differentiation states at relapse.

## A relapse-like epigenetic signature is present at diagnosis in relapse-fated AML

The finding that certain diagnosis clones display relapse-like epigenetic states raises the possibility that specific epigenetic signatures may be selected for during therapy. To further determine whether cellular subsets at diagnosis harbored relapse-like epigenetic states, we scored each cell from our single-cell ATAC-seq data using the relapse signature determined from our clonally stable bulk ATAC-seq cohort (*Figure 2f*), and evaluated if it was enriched in any diagnosis cells (*Figure 4e*). As expected, relapse cells in patients SU360, SU484, and SU892 were enriched for the relapse signature to a significantly greater degree than diagnosis cells. Notably, in these three cases, subpopulations of cells were identified at diagnosis that harbored the relapse signature at levels comparable to the relapse cells. These results suggest that intercellular epigenetic heterogeneity and evolution may be mechanisms contributing to relapsed AML.

## Single-cell ATAC-seq reveals relapsed AML cell state evolution

Next, we sought to determine how AML differentiation states changed at relapse at the single-cell level. We utilized latent semantic index projection to map AML scATAC-seq data onto a reference constructed from healthy hematopoietic cell scATAC data (*Figure 4—figure supplement 2a and b*). Each AML cell was classified based on the healthy hematopoietic cell type to which it mapped closest in epigenetic space according to this projection. We found that the differentiation status of each AML sample broadly reflected the results obtained from our analyses of bulk AML profiles (*Figure 4—figure supplement 2c–d*). However, single-cell mapping allowed for a significantly more granular identification of the differentiation status for all AML subpopulations in each sample. Significantly, the differentiation status as determined by the closest normal cell type changed substantially at relapse with cells either becoming more HSPC-like or mapping to a different hematopoietic lineage altogether. Consistent with our bulk data and with what we expected for a myeloid disease, patient SU484 mapped to the myeloid compartment at diagnosis, and at relapse became substantially more immature with most relapse cells becoming CMP/LMPP-like (*Figure 4—figure supplement 2e*). Patient SU892 diagnosis cells also largely mapped to the myeloid lineage, but the relapse cells interestingly appeared to project onto more dendritic/basophil and erythroid-like states, perhaps reflecting an overall more neomorphic relapsed cell state than could be mapped onto one cell-type (*Figure 4—figure supplement 2f*). Interestingly, case SU360 showed substantial epigenetic differences between diagnosis and relapse when comparing global AML profiles in both the bulk and single-cell data with an increase in the fraction of cells mapping to the DC/Basophil-like compartment (*Figure 4—figure supplement 2d*). Altogether, these data show that the epigenetic-based lineage status of AML subpopulations can change dramatically at relapse and that modulation of cell-type specific epigenetic programs is associated with relapse.

## Mitochondrial single-cell ATAC-seq reveals intracellular and convergent evolution in mutationally stable AML at relapse

Our initial single-cell ATAC-seq experiments provided evidence that there was substantial epigenetic change within, and possible selection of, AML subclones unrelated to somatic mutation change. To definitively identify the subclonal chromatin changes occurring within these subpopulations at relapse, we employed mitochondrial single-cell ATAC-seq (mtscATAC-seq), a technique which leverages genetic variants accumulating naturally within the mitochondrial genome as an endogenous barcode to track cell populations while simultaneously capturing their chromatin accessibility profile (*Lareau et al., 2021*). We performed mtscATAC-seq on the 10 X Chromium platform using FACS-purified

AML blasts from the four patients interrogated above. With this method, we were able to link AML subclones defined by mitochondrial mutations (henceforth referred to as 'mitoclones') between diagnosis and relapse, evaluate clonal dynamics, and identify epigenetic differences between mitoclones within and between timepoints (*Figure 5a*). Importantly, these mitoclones were identified agnostic of any somatic mutations previously characterized in these samples, allowing for the unbiased evaluation of subclonal dynamics in samples with no significant change in driver mutational profile at relapse. Additionally, the ability of this methodology to accurately identify mitochondrial variants and mitoclones as well as compare epigenetic differences between these clones has been previously validated (*Lareau et al., 2021*; *Rückert et al., 2022*).

The genomically stable AML cases evaluated in our study showed some level of subclonal heterogeneity in their respective mitoclones. In samples SU142, SU360, and SU892, 95% of the AML cells at diagnosis and relapse constituted 3–4 mitoclones (*Figure 5b*). Reconstruction of mitoclone evolution during therapy showed that at relapse, mitoclone dynamics were stable with no significant change in the size of mitochondrially defined clones (*Figure 5c*). Importantly, many of the mitoclones were epigenetically distinct from one another at diagnosis, indicating that different epigenetic programs were active across these populations at diagnosis (*Figure 5d*).

To determine how epigenetic profiles changed between individual mitoclones at relapse, we calculated the global epigenetic similarity between all diagnosis mitoclones and separately, calculated the epigenetic similarity between all relapse mitoclones. In the genomically stable samples, there was a significant change in the bulk AML epigenetic profile at relapse, with relapse mitoclones overall clustering separately from diagnosis mitoclones. Additionally, mitoclones at relapse were epigenetically more similar to each other than mitoclones at diagnosis in two of three stable samples evaluated (*Figure 5d*). When taken together, these two observations suggest that all relapse mitoclones evolve toward a distinct epigenetic state while also becoming epigenetically more similar to each other, demonstrating convergent epigenetic evolution in these populations.

We further evaluated epigenetic change across all genomic features between specific mitoclones. In SU360, for example, mitoclone 1 and mitoclone 2, clones with distinct evolutionary origins, showed similar patterns of accessibility change at relapse (*Figure 5—figure supplement 1a*). These data support the idea that single AML cells give rise to lineage-biased and epigenetically distinct subclones that may contribute to resistance and relapse independent of genomic alterations. In addition, intracellular epigenetic evolution can occur across AML subclones with convergent epigenetic evolution of distinct AML clones in patients with stable genomic profiles at relapse.

## Mitochondrial single-cell ATAC-seq reveals evolution of epigenetically distinct AML subpopulations at relapse

We further utilized mitochondrial tracing to profile the clonal and epigenetic profile of SU484, a case in which a FLT3 point mutation detected at diagnosis was lost and a FLT3-internal tandem duplication (ITD) was gained at relapse. While the stable samples profiled had relatively simple mitochondrial clonal heterogeneity, clonal structure was considerably more complex in SU484. We identified 12 mitoclones that underwent various levels of selection, elimination, and outgrowth, and which had distinctive epigenetic profiles at both diagnosis and relapse, reflecting a subclonal heterogeneity not evident through the investigation of somatic driver mutations and their dynamics (*Figure 5—figure supplement 1b–d*). Mitoclone 2, present at a frequency of 0.2% at diagnosis was selected, making up 36% of the AML cells at relapse, reflecting the selection of a clone likely associated with the FLT3-ITD driver mutation (*Figure 5—figure supplement 1c*). The small number of detectable mitoclone 2 cells at diagnosis were epigenetically distinct from other clones, suggesting that this clone contained a more favorable epigenetic state amenable to selection following therapy (*Figure 5—figure supplement 1d*). At relapse, mitoclone 2 cells became epigenetically distinct from mitoclone 2 cells at diagnosis, indicating that there was both selection for and evolution of this clone during therapy. On the other hand, several diagnosis mitoclones in SU484 appeared to undergo negative selection including mitoclones 1, 3, and 4 (*Figure 5—figure supplement 1c*). While these 'unfit' clones were distinct in their origins, they contained similar epigenetic profiles at diagnosis associated with their elimination at relapse (*Figure 5—figure supplement 1d*).

To further explore epigenetic differences between mitoclones, we mapped SU484 cells to the healthy hematopoietic reference and identified the closest normal cell type for different clones.

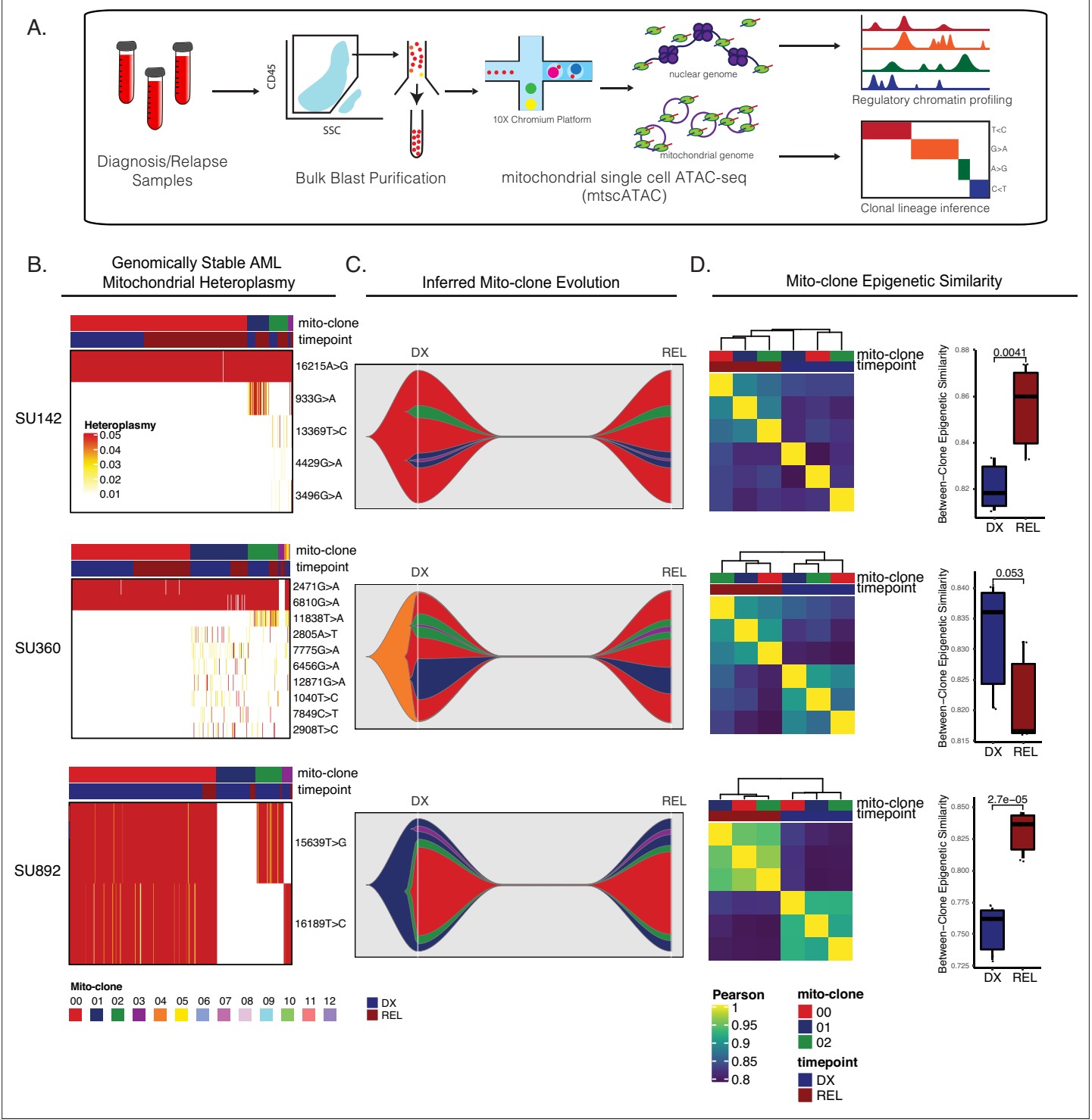

**Figure 5.** Mitochondrial-based clonal tracing paired with single-cell chromatin accessibility in stable AMLs. (**a**) Scheme for sample processing and analysis of mitochondrial single-cell ATAC-seq (mtscATAC-seq). (**b**) Heatmap of mitochondrial variant heteroplasmy values across all single cells for each genomically stable sample. Only variants passing filtering criteria are shown. Cells are ordered based on the cluster they are assigned to. (**c**) Fishplots of inferred mitochondrial clone evolution at diagnosis and relapse based on mitochondrial cluster frequencies in (**B**). (**d**) Heatmap of epigenetic similarity across all major mitoclones for each sample (left). Boxplot of inter-clone epigenetic similarity at diagnosis and relapse (means compared using unpaired student's t-test) (right).

The online version of this article includes the following figure supplement(s) for figure 5:

**Figure supplement 1.** Mitochondrial-based clonal tracing paired with single -cell chromatin accessibility in a genetically non-stable AML.

*Figure 5 continued on next page*

*Figure 5 continued*

**Figure supplement 2.** Clone-specific epigenetic relapse scores at diagnosis and relapse timepoints.

Strikingly, certain clone-defining mitochondrial variants were lineage biased. In SU484, one mitochondrial clone/variant was restricted to the myeloid compartment while others were distinctly progenitor-like (*Figure 5—figure supplement 1e*). Notably, mitoclone #2 that expanded at relapse was progenitor-like at diagnosis. In the stable samples with no mitoclone changes, mitoclones did not display lineage bias with diffuse mapping across multiple lineages in the hierarchy.

We additionally scored different mitoclones using our previously identified relapse signature and found that in all relapse samples, all clones displayed higher relapse-signature scores (*Figure 5—figure supplement 2*). In addition, in SU484, clone 0 and 2 which survived and expanded at relapse respectively had epigenetic states at diagnosis with higher relapse scores relative to clones that were eliminated during therapy. These data indicate that certain epigenetically distinct clones may be primed to relapse. Overall, these results demonstrate that epigenetic factors contribute to resistance and relapse in bulk AML and within epigenetically defined subclones that persist into relapse.

## Discussion

Relapsed AML remains a significant clinical challenge, and the mechanisms driving this disease state are incompletely understood. In this study, we found that many AML cases undergo no mutational evolution at relapse, suggesting alternative, non-genetic means that drive AML evolution during treatment and after remission. Our genotyping and chromatin accessibility analysis of paired diagnosis and relapsed AML samples indicates that genomic evolution and epigenetic evolution are two intertwined, yet independent mechanisms contributing to relapse, with specific epigenetic alterations occurring commonly across genetically stable samples. These findings are consistent with a prior study that identified diversity in cytosine methylation change at AML relapse, independent of genetic evolution (*Li et al., 2016*). We further found that epigenetic changes are not a reflection of increased leukemia stem cell proportion at relapse but represent alterations in regulatory sequences occurring across all subpopulations in the AML hierarchy. Single-cell ATAC-seq and analysis of mitoclones in diagnosis and relapse cell populations shows that convergent intracellular evolution occurs, with distinct mitoclones developing similar epigenetic features at relapse. This information further underscores the importance of the role of epigenetic factors in the clinical progression and pathogenesis of leukemia. Further and larger scale studies will need to be conducted to link specific mutational profiles with epigenetic alterations and clonal evolutionary patterns to understand the molecular mechanisms of AML relapse and guide future diagnostic tools and therapies.

Leukemia stem cells (LSCs) have long been thought of as a potential reservoir for relapse after clinical remission (*Dick, 2005*; *Thomas and Majeti, 2017*; *Jordan, 2007*). Our data indicates that epigenetic changes occurring in LSCs contribute significantly to the overall epigenetic evolution taking place in each AML, likely driving the observed resistance and relapse of these cases. Interestingly, the LSC-enriched cells we investigated exhibited epigenetic stability between the diagnosis and relapse, suggesting that while non-LSC epigenetic changes are derived from changes occurring in stem cells, the LSC-enriched cells may indeed represent a latent population contributing to eventual relapse (*Shlush et al., 2017*; *Parkin et al., 2017*). Our gene score data further suggests that these cells acquired inflammatory genetic programs at relapse, consistent with past reports of residual relapse-prone cells induced by chemotherapy (*Duy et al., 2021*). Further studies involving remission samples and epigenetic analysis of latent LSC populations across disease states could clarify the role of these cells in the initiation of AML relapse.

Single-cell mitochondrial ATAC-seq demonstrated that during chemotherapy, the AML as a whole evolves, with all populations changing their epigenetic state to occupy a distinct chromatin profile. At the same time, relapse mitoclones became epigenetically more similar to each other after chemotherapy in multiple samples, revealing a pattern of epigenetic convergent evolution. This is consistent with the idea of specific epigenetic traits and pathways being selected for during therapy, demonstrated by our bulk ATAC-seq data. These findings demonstrate that epigenetic evolution occurs in similar ways across genomically stable AMLs and that within these AMLs, epigenetic selection results in the convergence of global epigenetic states across clones.

The stable cases in our relapsed AML cohort exhibited a shared chromatin accessibility signature, suggesting the existence of a favorable epigenetic state contributing to eventual relapse which reflected activation of cell cycle gene pathways and DNA damage response. Based on this bulk ATAC-seq data, we hypothesized that epigenetic evolution could occur via different clonal mechanisms: either intercellular epigenetic evolution, wherein epigenetic cell clones with fixed chromatin states could be selected for at relapse, or intracellular epigenetic evolution, whereby cell populations across the entire AML adopt similar alterations in chromatin accessibility that permit survival and chemoresistance. Our single-cell ATAC sequencing data demonstrated that in some patients, epigenetically defined subpopulations of AML cells are selected for during therapy based on their epigenetic similarity to relapse epigenetic states, independent of genomic stability. In addition, our mitochondrial tracing data revealed that the mitoclones in the genetically stable AMLs were stable and did not demonstrate the selection of specific mitochondrial clones at relapse. We hypothesize that the selection of epigenetically defined clones is not detected by mitochondrial tracing since the AMLs analyzed in this study relapsed over the course of months and that there may not have been time for mitochondrial variants to accumulate in the epigenetically defined clones that were selected for at relapse in the stable samples.

Notably, we performed mitochondrial tracing in one genetically non-stable sample and found that there was substantial selection for and against multiple mitoclones in this sample. Although this represents a single AML, when considered alongside the stable samples, our data is consistent with the idea that mitochondrial variant heteroplasmy and dynamics capture AML evolution over longer periods of time and their stability is independent of the selection for epigenetic states or clones that occurs during shorter periods of time. In addition, the stability of mitoclones at relapse in the stable samples is consistent with the findings from our genetic studies that showed clonal stability, indicating that the samples we classified as stable in our study truly did not acquire or lose genetically defined clones during therapy and that our genetic studies were adequately powered to classify samples into clonal groups.

Several mechanisms could account for the epigenetic plasticity reported in many of the AML relapse patients from this study. Virtually all the samples queried here harbored mutations in epigenetic modifiers regulating chromatin, such as DNA methylation, histone methylation, or transcription factors such as RUNX1, reflecting the known landscape of epigenetic disruption by recurrent mutations in AML. It is possible that the altered chromatin state conferred by the loss of function of these regulators creates a permissive epigenetic state, allowing a 'flexibility' of the epigenome taken advantage of by cancer cells to grant an evolutionary growth or survival advantage, as has been previously postulated (*Corces and Corces, 2016*). Several studies in the past have highlighted the importance of the non-coding regulatory genome in leukemia, and cancer generally (*Li et al., 2020*; *Zhang and Meyerson, 2020*). We additionally hypothesize that since shorter time to relapse was associated with stable relapses, a lack of genomic evolution at relapse may result from a lack of time for mutations to accumulate and clonally expand. Alternatively, the shorter time to relapse in stable AMLs may be due to inherently more aggressive disease in these cases. The data reported here further reinforce the utility of chromatin accessibility as a readout for cell identity and epigenetic landscape. Future analysis of epigenetic mechanisms of relapsed AML will require the analysis of chromatin modifications, including DNA (hydroxy)methylation and histone marks, to specifically characterize the modification of gene regulatory sequences and specific pathways driving epigenetic evolution at different stages of leukemia and tumorigenesis.

The diversity of clonal evolutionary patterns characterized by our single-cell data highlights the challenge of developing novel therapeutics for, and curing, relapsed leukemia. Although we identify a broadly relapse-specific chromatin signature for mutationally stable samples, we note that there remains striking heterogeneity between patients in the dynamics by which various leukemic clones grow or recede at relapse. Our data also suggests mutational evolution and epigenetic evolution are connected, yet ultimately separate phenomena, as we observed several samples where one process appears to occur in the absence of the other. Moreover, we also identify several patients where there is little apparent clonal evolution, either via genetic variant marks or epigenetic configuration, begging the question of how these cases gain the ability to relapse. Additionally, other cell-extrinsic factors, interactions with the immune system and/or microenvironment, metabolic pathways, or other processes not made apparent through our analysis may have provided additional factors contributing

to relapse. Future efforts to detail the cellular mechanisms of relapse and identify therapeutic avenues will likely require analyzing patient-specific mutations or epigenome-specific chromatin features to combat relapsed cell clones. Altogether, this study demonstrates the heterogeneity and importance of epigenetic evolutionary mechanisms including convergent evolution in the relapse of AML.

## Methods
### Relapsed AML genomic meta-analysis
A PubMed literature search was performed to identify papers studying mutational profiles in relapsed AML samples. Studies were screened manually and included in the meta-analysis if paired AML samples from both diagnosis and relapse timepoints were profiled, next generation sequencing was utilized, and raw or processed sequencing as well as clinical data were publicly available. After inclusion and exclusion, a total of seven studies were identified which contained paired diagnosis-relapse genotyping data from a total of 312 AML patients.

Mutation and clinical data from all studies was standardized and collated using R for further analyses and patients were filtered out of the cohort if data was not available to evaluate mutation VAFs at both diagnosis and relapse. After filtering, the cohort consisted of 216 patients which were used in further analysis. Gene name, variant allele frequency, and genomic location were consolidated for all reported mutations for each study. Age, sex, ELN risk stratification, AML type (de novo vs secondary), days to relapse, and karyotype were consolidated for each patient when available. We utilized an approach previously described in multiple other studies to define the gain and loss of mutations at relapse (*Greif et al., 2018*; *Parkin et al., 2013*; *Rothenberg-Thurley et al., 2018*; *Stratmann et al., 2021*). Mutations were defined as 'gained' if the VAF went from below 0.05 to above 0.1 between diagnosis and relapse timepoints. Mutations were defined as 'lost' if the VAF went from above 0.1 to below 0.05 between diagnosis and relapse. All other mutations were classified as 'stable'. Importantly, all studies utilized in the meta-analysis had detection cutoffs at or below 0.05, allowing us to standardize the classification of mutations into gained, lost, and stable categories. For mutations with non-quantifiable VAFs (FLT3 ITD and NPM1 insertions), mutations were classified as either absent or present based on the analytical methods used in the associated studies.

The combined meta-analysis dataset was then utilized for downstream analyses. The 'oncoprint' mutation map was generated using custom scripts and the ComplexHeatmap package in R. The ternary plot for mutation visualization was generated using a modified oncoprint matrix data file (available in supplement) and the Ternary package in R. Relapse free survival between genomically 'stable' and 'unstable' cases was compared using Kaplan-Meier analysis; significance was determined using cox proportional hazards analysis using age and sex as covariates in the model (R packages survival and survminer).

### Human AML samples
Human AML samples were obtained and banked from patients at the Stanford Medical Center with informed consent, according to institutional review board (IRB)-approved protocols (Stanford IRB, 18329 and 6453). Patients were consented for sample collection and consent to publish was obtained at the time of sample collection. This study was performed per the IRBs using the sample bank. Thus, for this study, there was no direct contact with patients and no additional procedures. Mononuclear cells from each sample were isolated by Ficoll separation, resuspended in 90% FBS +10% DMSO, and cryopreserved in liquid nitrogen. All analyses conducted here on AML cells used freshly thawed cells.

### Cell sorting
Cell samples were first thawed and incubated at 37 °C with 200 U/mL DNase in IMDM +10% FBS. To enrich for CD34 + cells, magnetic bead separation was performed using MACS beads (Miltenyi Biotech) according to the manufacturer's protocol.

For cell staining and sorting, the following antibody cocktail was used with the sorting schemes shown in *Figures 2A, 3A and 5A* for respective cell type and library preparation:

- CD34-APC, clone 581, Biolegend, at 1:50 dilution.
- CD38-PE-Cy7, clone HB7, Biolegend, 1:25 dilution.
- CD19-PE-Cy5, clone H1B9, BD Biosciences, 1:50 dilution

- CD20-PE-Cy5, clone 2H7, BD Biosciences, 1:50 dilution
- CD3-APC-Cy7, clone SK7, BD Biosciences, 1:25 dilution
- CD99-FITC, clone TU12, BD Biosciences, 1:20 dilution
- TIM3-PE, clone 344823, R and D Systems, 1:20 dilution
- CD45-KromeOrange, clone J.33, Beckman Coulter at 1:25 dilution

Samples were sorted using a Becton Dickinson FACS Aria II. AML blasts were resuspended and kept in cold FACS buffer containing 1 ug/mL propidium iodide prior to and after sorting. Cells were then immediately prepared for downstream processing.

## Targeted genome sequencing

Targeted next generation sequencing (NGS) was performed using commercially available reagents according to the manufacturer's specifications. Briefly, purified genomic DNA was sheared using a Covaris S220 instrument to 250 bp fragments, end repair and adapter ligation was performed using the Celero DNA-Seq kit (Tecan Genomics, Redwood City, CA, USA) and hybrid capture was performed using the xGen AML Cancer Panel (IDT, Coralville, IA, USA). Libraries were sequenced on the Illumina HiSeq platform. Variant calling was performed using an established pipeline in the Majeti Lab as described below. Briefly, after quality control and read trimming, reads were aligned to the hg38 reference genome using BWA-MEM *Li and Durbin, 2009*, sorted, indexed and de-duplicated using SAMTools *Li et al., 2009* and Picard (http://broadinstitute.github.io/picard); RRID:SCR_006525, Version: 2.23.9, respectively. Variants were called using Mutect2 (*Cibulskis et al., 2013*), VarScan2 (*Koboldt et al., 2012*) and VarDict (*Lai et al., 2016*) in parallel. Only variants called by ≥ 2 callers were retained and annotated using SnpEff (*Cingolani et al., 2012*). Called variants were then used to determine major mutational shifts between disease states; 'major clone' mutations were defined as mutations having a detected average variant allele frequency of greater than 5%. Mutations were also categorized according to known disease relevance by sorting into three 'tiers': Tier 1 mutations known to play a role in AML and hematologic malignancies, Tier 2 mutations with known or potential onco-genic activity, and Tier 3, synonymous variants or variants of unknown significance. Tier 3 variants were filtered out prior to evaluating genomic evolution at relapse. All mutations were manually reviewed using IGV to visualize bam files for validation and verification of pathogenicity.

## Bulk ATAC-seq library preparation and sequencing

Sorted AML blasts were prepared for ATAC-seq as previously described (*Corces et al., 2017*). A total of 5000–50,000 cells were washed in cold FACS buffer and spun at 4 °C for 5 min at 500 rcf in a fix-angled centrifuge. Cell pellets were then resuspended in 50 μL of ATAC-seq resuspension buffer (RSB: 10 mM Tris-HCl pH 7.4, 10 mM NaCl, and 3 mM MgCl2 in water) with 0.1% NP40, 0.01% digitonin, and 0.1% Tween-20 and incubated on ice for three minutes. After lysis, 1 mL of ATAC-seq RSB with 0.1% Tween-20 was added and tubes were inverted six times to mix. Isolated nuclei were then spun at 4 °C for 10 min at 500 rcf in a fix-angled centrifuge. Supernatant was removed and nuclei were resuspended in 50 μL transposition mix (25 uL 2xTD buffer, 2.5 μL Tn5 transposase (100 nM final), 16.5 μL PBS, 0.5 μL 1% digitonin, 0.5 μL 10% Tween-20, and 5 L nucμlease-free water). Transposition reactions were incubated at 37 °C for 30 min in a thermomixer with shaking at 1000 rpm. Reactions were cleaned up using Qiagen MinElute Reaction Cleanup kits and processed as previously described. All libraries were amplified with a target concentration of 20 μl at 4 nM, which is equivalent to 80 femto-moles of product. All libraries were sequenced on an Illumina NextSeq with 75 bp paired end reads.

## Bulk ATAC-seq pre-processing, peak calling and merging, and count matrix generation

ATAC-sea data was processed based on workflows similar to *Corces et al., 2018*. First, read trimming and quality control was performed using trim_galore using default parameters. Reads were aligned to the GRCh38 reference genome using bowtie2 and the `--very-sensitive` option. Aligned reads were converted to bam format and sorted using SAMtools, and deduplicated using Picard MarkDuplicates. Reads were subsequently indexed and mitochondrial reads were removed using SAMTools. TagAlign bed files from technical replicate libraries were generated using bedtools, shifted to find ATAC-seq cut sites, and pooled. Peaks were called using MACS2 using p-value cut-off of 0.01 and peaks falling within black list regions (https://sites.google.com/site/anshulkundaje/projects/blacklists)

were removed. A consensus peak set was generated using methods from *Corces et al., 2018*. Briefly, called peaks were merged by merging all peak bed files and using an iterative removal process to resolve overlapping peaks. This approach resulted in a consensus set of 115,551 500 bp fixed-width peaks covering accessible regions represented in at least two or more samples. Count matrices were generated using the consensus peak set, and the number of Tn5 cut sites in each peak were calculated using the CountOverlaps function from the GenomicRanges package in R.

### Diagnosis vs relapse chromatin similarity

Global chromatin similarity between diagnosis and relapse samples for each patient was calculated by first averaging the peaks counts per million vector between replicates across all samples and taking the pearson-product moment correlation between diagnosis and relapse timepoints. This approach was utilized for both bulk AML ATAC-seq samples and LSC-enriched ATAC-seq samples.

### Gene accessibility scores

Accessibility scores were generated across all genes across all samples utilizing an approach previously described in *Granja et al., 2021* which was shown to estimate the expression levels of a gene more faithfully than other methods (*Granja et al., 2021*). Using the ArchR approach, a 5 kb tile matrix was utilized and for each gene, tiles falling within TSS-5kb to TTS were assigned a weight of $1+e^{-1}$, and flanking tiles were assigned a weight using an exponential decay model of $e^{-abs(distance)/5000}+e^{-1}$. Tiles were assigned a weight of 0 if they ran into adjacent genes or if they were >20,000 bp from the gene. Tn5 insertions were normalized per tile, and summed per gene to generate a gene accessibility score. These scores were then used to generate a gene accessibility score matrix which was utilized in subsequent analyses.

### Differential accessibility

Differential accessibility analysis was performed using the Tn5 cut sites per peak matrix or gene accessibility scores matrix. Raw count matrices were utilized as input to the DESeq2 model for this purpose. The DESeq model was used with default parameters to normalize the data and perform differential accessibility analysis across all features. Differential accessibility was performed across stable samples between diagnosis and relapse timepoints using patient labels as covariates to control for between patient differences.

### Chromatin accessibility gene set enrichment analysis

Gene set enrichment analysis was performed using the GSEA Mac App v4.2.3 (Broad Institute, Inc) and the fgsea R package using a gene accessibility score matrix described above pre-ranked according to relapse vs diagnosis ATAC gene accessibility log fold change. Results were filtered by p-adjusted <0.05.

### Bulk projection to healthy hematopoietic reference

To map bulk AML ATACseq samples to a healthy single cell reference and determine closest normal cell types, we utilized an approach from *Granja et al., 2019*. This approach involved the generation of pseudo-single cell ATACseq samples from bulk samples and the utilization of latent semantic indexing to project these pseudo-single cell libraries to the healthy reference. Briefly, pseudo-single cell samples were generated for each bulk ATAC sample by sampling reads to generate 250 pseudo-single cells per sample. These pseudo-single cell samples were then subjected to LSI and projection to the healthy hematopoietic reference. For full details, see 'Projection to healthy hematopoietic reference' below, and *Granja et al., 2019*.

### Leukemia stem cell chromatin accessibility analysis

ATACseq data from sorted LSC-enriched cell populations was processed in a similar way as the bulk AML cells (described above).

### 10X single-cell ATAC-seq

Sorted AML blasts were prepared with 10 X Genomics Chromium Single Cell ATAC Kit v1.1 according to the manufacturer's specifications. Samples were uniquely barcoded and quantified using an Agilent

BioAnalyzer 2100 or a KAPA qPCR quantification kit. Sample libraries were loaded onto an Illumina NovaSeq and sequenced with 50x8 × 16 x50 read configuration with an average of 25,000 paired end reads per single cell.

## scATAC-seq analysis

scATAC-seq data was processed using a combination of cellranger, ArchR, and custom analysis scripts described below. All analysis was performed in R version 4.0.5 unless otherwise specified.

## Data processing, dimensional reduction, and data filtering

scATAC fastq files were processed using the cellranger pipeline with standard parameters. Fragment files from cellranger were processed with the ArchR software to generate Arrow files using the create-ArrowFiles function for each single-cell sample which were subsequently stored locally for downstream processing. Using ArchR, doublet scores were assigned to each cell and cells deemed to be doublets by ArchR were removed from all samples. Next, latent semantic indexing, batch correction, and clustering were performed using the ArchR functions addIterativeLSI, addHarmony, and addClusters functions, respectively.

To visualize the accessibility of specific genes across cells, we utilized the ArchR function addGene-ScoreMatrix to generate a gene accessibility score matrix for each sample. These matrices were used in downstream analyses to visualize and quantify gene-specific chromatin accessibility. Samples in the study were visualized using UMAP dimensional reduction which was performed using the ArchR addUMAP function and default parameters. Upon visualization of clusters, we observed that for three of four samples, there existed a low percentage of cells that clustered separately from the bulk of AML cells. We hypothesized that these differences were biologically relevant or were caused by residual healthy immune cells captured and sequenced during sample preparation. To test this, we quantified and visualized the accessibility of genes shown to be highly expressed in T cell (CD89, TLR4, GZMA, CD247), B cell (TCL1, CD37), monocyte (CD209), and other cell subsets. Clusters that exhibited high accessibility of these genes which were overall epigenetically dissimilar to the bulk of the sample were filtered and excluded from downstream analysis as they were deemed to be residual contaminating healthy immune cells.

## Projection to healthy hematopoietic reference

In order to identify the closest normal celltypes associated with cells in the AML single cell ATACseq data, we used an approach from *Granja et al., 2019* which involves Latent Semantic Indexing (LSI) and projection of the sparse single cell data onto a single cell ATACseq manifold derived from healthy hematopoietic cell types (*Granja et al., 2019*). This approach ultimately results in a low dimensional representation of both the healthy hematopoietic reference data as well as the AML single cell ATAC data through which the projection can be visualized and the closest normal cell type for each AML cell can be calculated. For full details of analysis, please see the codebase for this manuscript or *Granja et al., 2019*.

Following the application of LSI, dimensional reduction was further performed using the UMAP algorithm as described in Granja et al. The output of the UMAP projection was used to visualize the AML cells alongside the healthy hematopoietic scATAC reference. To classify AML cells based on their nearest healthy hematopoietic neighbors in epigenetic space, we used the LSI dimensional reduction output from both healthy and diseased cell states and classified each disease cell based on the top 10 healthy cells that it was closest to in LSI space using the knn function in the class package in R. These classifications were then used for visualization and quantification of the overall differentiation status and closest normal cell types for each cell and overall AML for each sample.

## Relapse chromatin score

We generated a 'relapse chromatin score' for each cell evaluated using scATACseq in the study to determine how 'relapse-like' each cell was and whether there was variability in relapse-like states within samples. To generate a relapse score for each cell, we utilized the addFeatureCounts function from ArchR to sum the ATAC enrichment across the top 500 genes in our relapse signature derived from the bulk stable samples (analysis shown in *Figure 2*). We then subtracted from this value the sum of the bottom 500 genes from the same relapse signature. This resulted in a single numerical value

per cell with higher values indicating that the cell contained an epigenetic state more similar to bulk relapse samples and less similar to diagnosis relapse samples. We then visualized and quantified these values across all cells and samples evaluated in the study.

## mtscATAC-seq
### Library preparation and sequencing
Mitochondrial single-cell ATAC-seq was performed as previously described (*Lareau et al., 2021*). Modifications were made to the 10 X Single-cell ATAC-seq protocol to accommodate capture of mitochondrial reads: briefly, freshly sorted AML blast cells were washed in PBS and fixed in 1% formaldehyde in PBS for 10 min at room temperature. Fixation was quenched with glycine to a final concentration 0.125 M, then washed twice in PBS and spun at 400 × $g$ for 5 min in a fixed-angle centrifuge at 4 °C. Cells were then treated with a gentle lysis buffer consisting of 10 mM Tris-HCl pH 7.4, 10 mM NaCl, 3 mM MgCl$_2$, 0.1% NP40, and 1% BSA for 3 min on ice, then 1 ml of chilled wash buffer (10 mM Tris-HCl pH 7.4, 10 mM NaCl, 3 mM MgCl$_2$, 1% BSA) and mixed by inversion before centrifugation. Cells were then processed with 10 X Chromium Single Cell ATAC Kit v1.1 according to the manufacturer's protocol. Samples were uniquely barcoded and quantified using an Agilent BioAnalyzer 2100. Sample libraries were loaded onto an Illumina NovaSeq and sequenced with 50x8 × 16 x50 read configuration with an average of 25,000 paired end reads per single cell.

### mt-scATAC-seq analysis mgatk variant heteroplasmy identification
Raw sequencing reads were demultiplexed using CellRanger-ATAC mkfastq and aligned to the host reference genome using CellRanger-ATAC v2.0 with default settings. Somatic mitochondrial DNA mutations were called using the mgatk software v0.6.2 using the `tenx` mode and the default parameters. Somatic mutations used for mitoclone analyses were determined from the default output of the mgatk workflow, including the variantStats file and subsequently clustering the per-mutation, per-cell heteroplasmy matrix.

## Clone and evolution hierarchy identification
Mitochondrial clones were identified as previously described in *Lareau et al., 2021*. First, mitochondrial variants were filtered based on heuristics previously described in Lareau et al. Variants were deemed high quality if the number of cells the variant was detected in was greater than 3, strand correlation during sequencing was greater than 65%, and mean coverage at the locus was greater than 10 reads. The variants which passed filtering were manually extracted and used in subsequent steps. Next, these variants were read into a variant by cell matrix containing heteroplasmy values for each variant in each cell. This matrix was then binarized and the *seuratSNN* function from the R package *Seurat* was used to cluster the matrix and identify unsupervised clusters of cells grouped by their mitochondrial variants. Clustering resolution was manually selected as previously described. The average heteroplasmy for each variant in each cluster was inspected to verify that clusters were not primarily driven by low heteroplasmy variants. The clonal hierarchy for each sample was determined manually by building a clonal evolution tree based on shared and exclusive mitochondrial variants across all clusters for each sample. The resulting hierarchy and corresponding mito-clone frequencies were plotted using the *fishplot* package in R.

## Mitoclone projection
Cells were projected to the healthy hematopoietic reference as described above. Mitochondrial variants were then mapped to the projection plot to display the differentiation status of cells containing various mitochondrial variants.

## Clone chromatin similarity
Clone similarity was calculated by generating a pseudo-bulk ATACseq sample for each cluster and then taking the pearson-product moment correlation across all peaks between pairs of clusters. This process was performed between all diagnosis clusters and all relapse clusters independently, and the resulting inter-time point similarities were compared between diagnosis and relapse for each sample.

## Acknowledgements

KAN acknowledges support by the Stanford Cancer Biology Graduate Program's NIH T32 Training Grant. AA was supported by the ASH HONORS award. TK is a Special Fellow of The Leukemia & Lymphoma Society. This work was supported by NIH Grants 1R01CA251331 (RM), HG012579 (CAL), HG012076 (CAL, MRC, ATS). Additional support was provided by the Stanford Ludwig Center for Cancer Stem Cell Research and Medicine (RM) This work was supported by a Lloyd J Old STAR Award from the Cancer Research Institute (ATS) and an ASH Scholar Award from the American Society of Hematology (ATS). We would like to thank all the members of the Majeti Lab for helpful comments and discussion, as well as members of Howard Chang's laboratory for scientific input. We thank the Flow Cytometry Core at the Stanford Institute for Stem Cell Biology and Regenerative Medicine for flow cytometry equipment, maintenance, training, and discussion. We would also like to acknowledge the Stanford Functional Genomics Facility for sequencing equipment and training for single-cell studies. We additionally thank the Stanford Cancer Biology Graduate Program directors and program administration for support and input throughout this project. Finally, we would like to thank the patients and their families for their contribution of samples for this study.

## Additional information

### Competing interests

Caleb Lareau: consultant for Cartography Biosciences. Ansuman T Satpathy: founder of Immunai and Cartography Biosciences; receives research funding from Allogene Therapeutics and Merck Research Laboratories. Ravindra Majeti: on the Advisory Boards of Kodikaz Therapeutic Solutions, Orbital Therapeutics, and is an inventor on a number of patents related to CD47 cancer immunotherapy licensed to Gilead Sciences. Co-founder and equity holder of Pheast Therapeutics, MyeloGene, and Orbital Therapeutics. The other authors declare that no competing interests exist.

### Funding

| Funder | Grant reference number | Author |
| --- | --- | --- |
| American Society of Hematology | | Armon Azizi |
| Leukemia and Lymphoma Society | | Thomas Koehnke |
| National Institutes of Health | | Caleb Lareau<br>M Ryan Corces<br>Ansuman T Satpathy<br>Ravindra Majeti |
| National Institutes of Health | 1R01CA251331 | Ravindra Majeti |
| National Institutes of Health | HG012579 | Caleb Lareau |
| National Institutes of Health | HG012076 | Caleb Lareau<br>M Ryan Corces<br>Ansuman T Satpathy |

The funders had no role in study design, data collection and interpretation, or the decision to submit the work for publication.

### Author contributions

Kevin Nuno, Conceptualization, Resources, Data curation, Validation, Investigation, Methodology, Writing – original draft, Writing – review and editing; Armon Azizi, Conceptualization, Resources, Data curation, Software, Formal analysis, Investigation, Visualization, Methodology, Writing – original draft, Writing – review and editing; Thomas Koehnke, Data curation, Formal analysis; Caleb Lareau, Software, Formal analysis; Asiri Ediriwickrema, Formal analysis; M Ryan Corces, Software,

Investigation; Ansuman T Satpathy, Supervision; Ravindra Majeti, Conceptualization, Supervision, Funding acquisition, Investigation, Writing – original draft, Project administration, Writing – review and editing

## Author ORCIDs
Kevin Nuno  http://orcid.org/0000-0002-2677-1319
Armon Azizi  http://orcid.org/0000-0002-1353-2060
Caleb Lareau  http://orcid.org/0000-0003-4179-4807
Ravindra Majeti  https://orcid.org/0000-0002-5814-0984

## Ethics
Human AML samples were obtained and banked from patients at the Stanford Medical Center with informed consent, according to institutional review board (IRB)-approved protocols (Stanford IRB, 18329 and 6453). Patients were consented for sample collection and consent to publish was obtained at the time of sample collection. This study was performed per the IRBs using the sample bank. Thus, for this study, there was no direct contact with patients and no additional procedures.

## Decision letter and Author response
Decision letter https://doi.org/10.7554/eLife.93019.sa1
Author response https://doi.org/10.7554/eLife.93019.sa2

---

# Additional files

## Supplementary files
• Supplementary file 1. Supplementary meta-analysis, clinical, treatment, and genotyping data. (a) List of all publicly available studies utilized in the meta-analysis of relapsed AML genomics. (b) Clinical, treatment, and karyotype information for all patients in the study. (c) Summary of all genotyping information for all paired diagnosis and relapse samples in the study. (d) Detailed genotyping information for all paired diagnosis and relapse samples in the study. (e) Bulk ATAC-seq gene score differential accessibility results from stable AML analysis.

• MDAR checklist

## Data availability
All sequencing and processed data are uploaded to GEO accession GSE256495. Code used for data analysis for all figures in this study is available at (https://github.com/armonazizi/Azizi_Nuno_AML_Relapse_Chromatin, copy archived at *Azizi, 2024*).

The following dataset was generated:

| Author(s) | Year | Dataset title | Dataset URL | Database and Identifier |
|---|---|---|---|---|
| Nuno KA, Azizi A, Köhnke T, Lareau CA, Ediwirickrema A, Corces MR, Satpathy AT, Majeti R | 2024 | Convergent Epigenetic Evolution Drives Relapse in Acute Myeloid Leukemia | https://www.ncbi.nlm.nih.gov/geo/query/acc.cgi?acc=GSE256495 | NCBI Gene Expression Omnibus, GSE256495 |

The following previously published datasets were used:

| Author(s) | Year | Dataset title | Dataset URL | Database and Identifier |
|---|---|---|---|---|
| Buenrostro J | 2016 | ATAC-seq data | https://www.ncbi.nlm.nih.gov/geo/query/acc.cgi?acc=GSE74912 | NCBI Gene Expression Omnibus, GSE74912 |
| Granja JM | 2019 | Single-cell, multi-omic analysis identifies regulatory programs in mixed phenotype acute leukemia | https://www.ncbi.nlm.nih.gov/geo/query/acc.cgi?acc=GSE139369 | NCBI Gene Expression Omnibus, GSE139369 |

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
