## [Editor Report]

The authors show convincingly that most relapses in AML occur without changes in driver mutations. By using ATAC-seq in matched diagnosis and relapsed samples, they show that epigeneticc mechanisms drive relapse in a large proportion of cases. These findings are of translational importance and are based on rigorous analysis of large set of primary samples.

---

## [Decision Letter]

**Decision letter after peer review:**

Thank you for submitting your article "Convergent Epigenetic Evolution Drives Relapse in Acute Myeloid Leukemia" for consideration by *eLife*. Your article has been reviewed by 3 peer reviewers, one of whom is a member of our Board of Reviewing Editors, and the evaluation has been overseen by Tony Ng as the Senior Editor.

Essential revisions (for the authors):

1) Rev 1: Analyzing whether certain mutational groups are associated with specific epigenetic findings as seen by ATAC-seq?

2) Rev 2: Address points brought by reviewer 2: unclear why only a subset of AML cases shows stable clonal patterns.- can the authors speculate?

*Reviewer #1 (Recommendations for the authors):*

The authors are commended for performing ATAC-seq on a large number of matched diagnosis and relapse samples. The chromatic accessibility maps show epigenetic changes at relapse and offer a resource for the community. The findings that most relapses are not caused by large mutational changes are also novel and of great interest to hematology researchers.

The authors can enhance the findings by providing a more detailed analysis of transcription factor changes at relapse that can be inferred by analysis of ATAC seq data.

Further improvements can be made by analyzing whether certain mutational groups are associated with specific epigenetic findings as seen by ATAC-seq.

*Reviewer #2 (Recommendations for the authors):*

It remains unclear why only a subset of AML cases shows stable clonal patterns. Several important pieces of information would be important to include in this regard. First, for the relapse cases, the authors should clarify when the relapse occurred and whether the patient was recently treated with chemotherapy. One could imagine that chemotherapy could induce similar patterns of chromatin accessibility in AML, but that late relapses might not exhibit such "convergent evolution" since the epigenetic selection pressures imposed by chemotherapy would be in the distant past. Second, it is important for the authors to fully describe the impact of alterations in chromatin stability in unstable clone cases to determine if there is any evidence for epigenetic evolutionary convergence in these cases, common alterations in molecular pathways, de-differentiation, or adoption of more LSC-like states. Third, the authors should provide some type of orthogonal evidence to support claims regarding biological differences between diagnosis and relapse blasts, especially concerning alterations in differentiation, self-renewal, and alterations in cell cycle-related pathways. For example, the authors could assess differentiation status or cell cycle status by blast immunophenotype and size, respectively. Finally, the authors should be circumspect when they call sorted blasts "LSCs" based CD99+TIM3+ expression. As stated by the authors, this strategy allows distinction of blasts from non-leukemic residual HSCs, but does not necessarily allow significant enrichment of LSCs, even in the CD34+CD38- compartment since the Dick and Carroll groups previously showed LSCs can express all combinations of these markers.

---

## [Author Response]

Essential revisions (for the authors):1) Rev 1: Analyzing whether certain mutational groups are associated with specific epigenetic findings as seen by ATAC-seq?

We thank the reviewer for raising this important point, as epigenetic changes at relapse specific to mutational groups have not been evaluated in the prior literature. Due to the limited number of samples and varied mutations across patients, there is a low number of samples in each mutational group in our cohort. Despite this limitation, we performed an analysis to identify mutation-specific epigenetic changes occurring at relapse by identifying chromatin changes specific to AMLs that had specific mutations present at both diagnosis and relapse. Inclusion criteria for each mutation in this analysis were that there were more than 5 samples with the mutation present at both diagnosis and relapse. Based on these criteria, mutations that were analyzed included FLT3, NPM1, TET2, and DNMT3A.

This analysis showed that for FLT3, NPM1, and DNMT3A, after multiple hypothesis testing correction, there were no chromatin loci that were significantly differentially changed at relapse between mutant and wildtype groups (Author response image 1). For TET2 mutant AMLs, after multiple hypothesis testing correction, only 5 gene loci displayed significantly different change at relapse in mutant samples compared to wildtype samples. LINC01476, PITPNM3, SPSB1, CORO2B, and TMEM202 loci were significantly more (Author response image 1). We attribute these predominantly negative findings to the low number of samples in each mutation group in our cohort. To fully address the reviewer’s comments and concerns, we have added points in our manuscript specifically stating that we did not find significant differences in chromatin evolution patterns in the mutational groups analyzed.

**Author response image 1. sa2fig1:** Volcano plots of differential analysis between mutant vs wildtype of relapse vs diagnosis fold change. Mutations analyzed include (A) FLT3, (B) DNMT3A, (C) NPM1, (D) TET2. Volcano plots are displayed with unadjusted (left) and adjusted (right) p values. In all analyses but one, p-value adjustment resulted in no significant gene loci chromatin evolution differences. In TET2-mutant samples, 5 loci remained differential after multiple-hypothesis testing correction.

2) Rev 2: Address points brought by reviewer 2: unclear why only a subset of AML cases shows stable clonal patterns.- can the authors speculate?

Thank you for raising this important point and question. While there are some analyses which were included in the submitted manuscript aimed at addressing the reviewer’s point, we appreciate that we did not speculate deeply about why only a subset of cases showed stable clonal patterns in the manuscript text. In the study, we initially hypothesized that clonal stability may be associated with or explained by the time from complete remission (CR) to relapse. Indeed, our analyses of both the metanalysis of genomic data as well as our own samples, demonstrate that AMLs with stable clonal patterns relapse more quickly when compared to AMLs with unstable clonality.

Based on these results, we speculate that the observed variation in clonal stability may be explained by 2 hypotheses.

Shorter times to relapse (TTR) might result in a higher likelihood that an AML is clonally stable. This may be related to the depth of clinical remission achieved and level of MRD, which is likely related to multiple factors including patient-specific, treatment-specific, and disease-specific features. We hypothesize that in cases where TTR is shorter, there is less time and therefore less opportunity for mutations to accumulate and subclones to expand into relapse. Conversely, in cases where TTR is longer, there is more time for mutations to accumulate in subclones and for these clones, or subclones lacking diagnosis mutations to expand, resulting in gained or lost mutations at relapse.

AMLs with stable clonality may be primed to relapse more quickly due to their underlying biology. Our analysis of differential chromatin accessibility at relapse in stable AMLs demonstrates that cell cycle and metabolic pathways are specifically upregulated in AMLs with stable clonality. These results suggest that there may be inherent biology specific to stable AMLs resulting in shorter TTR. These cases may represent a specific subtype of AMLs defined by a lack of genomic change and specific chromatin changes at relapse.

While we believe that both of these hypotheses are possible based on our results, we unfortunately lack the data or means to prove or disprove either conjecture. To further validate these hypotheses, a larger cohort of prospectively acquired diagnosis and relapse samples, stratified by their TTR would be required. However, to address the reviewer’s comment, we have further integrated these hypotheses into the Discussion section of the manuscript.